# Reversible Redox Property of Co(III) in Amorphous Co-Doped SiO_2_/γ-Al_2_O_3_ Layered Composites

**DOI:** 10.3390/ma13235345

**Published:** 2020-11-25

**Authors:** Shotaro Tada, Shota Saito, Akito Mori, Hideki Mizuno, Shiori Ando, Toru Asaka, Yusuke Daiko, Sawao Honda, Samuel Bernard, Yuji Iwamoto

**Affiliations:** 1Department of Life Science and Applied Chemistry, Graduate School of Engineering, Nagoya Institute of Technology, Gokiso-cho, Showa-ku, Nagoya 466-8555, Japan; s.tada.341@stn.nitech.ac.jp (S.T.); shota.saito@cgco.co.jp (S.S.); a_mori_10029@murata.com (A.M.); m.hideki0318@gmail.com (H.M.); shiori786@outlook.jp (S.A.); asaka.toru@nitech.ac.jp (T.A.); daiko.yusuke@nitech.ac.jp (Y.D.); honda@nitech.ac.jp (S.H.); 2Centre Européen de la Céramique, University of Limoges, 12 Rue Atlantis, 87068 Limoges, France; samuel.bernard@unilim.fr

**Keywords:** Co(III), aluminosilicate, reversible Co(III)/(II) redox property, membrane, H_2_-triggered chemical valve property

## Abstract

This paper reports on a unique reversible reducing and oxidizing (redox) property of Co(III) in Co-doped amorphous SiO_2_/γ-Al_2_O_3_ composites. The Fenton reaction during the H_2_O_2_-catalyzed sol–gel synthesis utilized in this study lead to the partial formation of Co(III) in addition to Co(II) within the composites. High-resolution transmission electron microscopy (HRTEM) and high-angle annular dark-field scanning transmission electron microscopy (HAADF-STEM) analyses for the composite powder sample with a composition of Al:Si:Co = 85:10:5 showed the amorphous state of the Co-doped SiO_2_ that modified γ-Al_2_O_3_ nanocrystalline surfaces. In situ X-ray absorption fine structure (XAFS) spectroscopic analysis suggested reversible redox reactions of Co species in the composite powder sample during heat-treatment under H_2_ at 500 °C followed by subsequent cooling to RT under Ar. Further analyses by in situ IR spectroscopy combined with cyclic temperature programmed reduction/desorption (TPR/TPD) measurements and X-ray photoelectron spectroscopic (XPS) analysis revealed that the alternating Co(III)/(II) redox reactions were associated with OH formation (hydrogenation)-deformation (dehydrogenation) of the amorphous aluminosilicate matrix formed in situ at the SiO_2_/γ-Al_2_O_3_ hetero interface, and the redox reactions were governed by the H_2_ partial pressure at 250–500 °C. As a result, a supported mesoporous γ-Al_2_O_3_/Co-doped amorphous SiO_2_/mesoporous γ-Al_2_O_3_ three-layered composite membrane exhibited an H_2_-triggered chemical valve property: mesopores under H_2_ flow (open) and micropores under He flow (closure) at 300–500 °C.

## 1. Introduction

Microporous amorphous silica (SiO_2_)-based membranes exhibit excellent hydrogen (H_2_)-selectivity and higher thermal stability compared to organic polymer membranes [1,2,3,4,5]. The SiO_2_ membrane component is also attractive for developing novel mixed matrix membranes combined with a polymer membrane component such as polysulfone [6], which shows better membrane performance with improved mechanical properties. Moreover, SiO_2_ membranes can be further modified as SiO_2_-based organic-inorganic hybrid membranes for other applications such as liquid-phase separation [7].

It has been recognized that high H_2_-selectivity can be achieved by the molecular sieving property of amorphous SiO_2_ with micro-pores of approximately 0.3 nm in size [3,4,5]. The microporosity can be formed in situ and is related to the highly branched fractal dimensions having a large amount of hydroxyl (Si–O–H) groups [8]. Accordingly, microporous SiO_2_ membranes are moisture sensitive [9] and lack hydrothermal stability [10]. 

Recently, it has been reported that doping amorphous SiO_2_ with metal (M) cations (M = Zr [11], Ni [12], Co [13,14,15]) improved the thermal or hydrothermal stability of sol–gel-derived SiO_2_ membranes. Among the dopants, Co was found to be effective at enhancing H_2_ permeance at 500 °C [16]. Nanostructure observation of the Co-doped SiO_2_-based membranes revealed that CoO and Co_3_O_4_ nanoparticles of approximately 5–20 nm in size were distributed within the SiO_2_ matrix [16]. More recently, SiO_2_-based membranes synthesized from Co-modified ethoxy group-functionalized polysiloxane have been reported to show a unique reversible gas molecular sieving property above 300 °C [17]. The reversible gas molecular sieving property was suggested to be governed by alternating volume shrinkage/expansion due to the reversible change in the reducing/oxidizing (redox) state of the Co oxide particles (Co(OH)_2_ and CoO/Co_3_O_4_) finely distributed within the polymer-derived SiO_2_ matrix [17]. These results reveal that the divalent(II) or trivalent(III) Co cations that remained under H_2_ atmosphere contributed to the enhanced H_2_ permeance at 300–500 °C.

On the other hand, reactions between H_2_ and various metal cations (M^m+^ = Ni^2+^ [18], Fe^3+^ [19], Ce^4+^ [19], Sn^4+^ [19], V^+5^ [20], Cu^+1^ [21], Ag^+^ [21]) in SiO_2_ glass have been reported: The cations are reduced to lower valence state, whereas protons in the form of hydroxyl are introduced into SiO_2_ glasses,
2(Si–O–Si)*_n_*–O^−^ + 2M^m+^ + H_2_ → 2(Si–O–Si)*_n_*–OH + 2M^(m−1)+^(1)
2(Si–O–Si)*_n_*–O^−^ + Sn^4+^ + H_2_ → 2(Si–O–Si)*_n_*–OH + Sn^2+^(2)

Analogous reactions have been reported for Cu^2+^ in zeolite derivatives [22], and other metal cations (M^m+^ = Eu^3+^ [23,24,25,26], Mn^3+^ [25], Ce^4+^ [27]) in Al_2_O_3_–SiO_2_ glasses,
2(Si–O–Al)*_n_*–O^−^ + 2M^m+^ + H_2_ → 2(Si–O–Al)*_n_*–OH + 2M^(m−1)+^(3)

In this study, Co cation-doped SiO_2_/γ-Al_2_O_3_ composite materials were synthesized; then, the reversible redox properties of Co cations around the hetero interface in the composites were intensively studied. First, a Co cation-doped SiO_2_ thin film was deposited on a mesoporous γ-Al_2_O_3_ layer and an equilibrium chemical composition of the Al, Si, and Co within the mesoporous γ-Al_2_O_3_ layer was determined. Then, redox properties of the Co cations in the Co-doped SiO_2_/γ-Al_2_O_3_ composite as powders with a defined equilibrium chemical composition were investigated by cyclic heat treatment at 300–500 °C under alternative flow gas change of H_2_ and Ar. Moreover, a unique H_2_-triggered chemical valve property of the γ-Al_2_O_3_/Co-doped SiO_2_/γ-Al_2_O_3_ three-layered composite membrane is discussed based on the unique reversible redox reactions of the Co cation for the purpose of developing novel hydrogen production, storage, and transportation systems essential for the establishment of a hydrogen-based society.

## 2. Experimental Procedures 

### 2.1. Sample Synthesis

Tetraethoxysilane (Si(OC_2_H_5_)_4_, purity > 99.0%), aluminum nitrate nonahydrate (Al(NO_3_)_3_·9H_2_O, purity 98.0%), cobalt(II) nitrate hexahydrate (Co(NO_3_)_2_·6H_2_O, purity 98.0%), ethanol (C_2_H_5_OH, purity > 99.5%), and hydrogen peroxide (H_2_O_2_, 30% aq.) were purchased from Kishida Chemical Co. Ltd., Osaka, Japan. 

In this study, for the sample syntheses of Co^−^-doped SiO_2_/γ-Al_2_O_3_ composites and Co^−^-doped aluminosilicate, Co^−^-doped SiO_2_ precursor solutions with Co/Si = 1/8, 1/4, and 1/2 were prepared according to a previously reported procedure [16]. Here, as an illustration, we describe the preparation of the Co-doped SiO_2_ sol with Co/Si = 1/8: A 10 mL round-bottom flask was charged with Co(NO_3_)_2_·6H_2_O (0.343 g) and C_2_H_5_OH (4.2 mL). To this solution, 1.92 mL (8 mmol) of Si(OC_2_H_5_)_4_ and H_2_O_2_ (2.3 mL, 30% aq) were added and maintained at RT with stirring for 1 h, then heated at 60 °C for 2 h. 

Prior to the composite sample syntheses, the Co/Si ratio of the precursor-derived Co-doped SiO_2_ was examined as follows: The Co-doped SiO_2_ sols with different Co/Si ratios were heated with stirring to 80 °C and kept at this temperature overnight to give dried Co-doped SiO_2_ gel powder samples. The dried gel powder samples were placed on a quartz boat in a quartz tube furnace (Model ARF-50KC, Asahi Rika Seisakusho Co. Ltd., Chiba, Japan), and heat-treated in air at 600 °C for 3 h with a heating/cooling rate of 20 °C·min^−1^ to afford Co-doped SiO_2_ powder samples. Then, elemental analyses were performed for Si and Co (ICP spectrometry, Model ICPS-7510, Shimazu Co., Kyoto, Japan), and carbon (C, non-dispersive infrared method, Model CS844, LECO Co., St Joseph, MI, USA). The residual carbon in the powder samples were negligible (<0.1%), and as shown in Appendix A, we confirmed that the Co/Si ratios in the range of 1/8 to 1/2 were well controlled through the present sol–gel route using H_2_O_2_.

#### 2.1.1. Co-Doped SiO_2_/γ-Al_2_O_3_ Layered Composite Sample

Boehmite (γ-AlOOH) coating solution was prepared using the previously reported procedure [28] and spin-coated on a SiO_2_ substrate (10 mm × 10 mm × 3 mm, Meijo science Co. Inc., Nagoya, Japan) at 3000 rpm using a spin coater (Model, MS-A100, Mikasa Co. Ltd., Tokyo, Japan). The γ-AlOOH-coated SiO_2_ substrate was placed on an Al_2_O_3_ plate and loaded in an electric furnace (Model FUW220PA, Advantec Toyo Kaisha Ltd., Tokyo, Japan) then heat-treated in air at 600 °C for 3 h with a heating/cooling rate of 100 °C·h^−1^ to afford the formation of a thin layer of γ-Al_2_O_3_ over the SiO_2_ substrate. Subsequently, Co-doped SiO_2_ sol with Co/Si = 1/8 was spin-coated on the γ-Al_2_O_3_ layer on the SiO_2_ substrate by using the same manner as described above. The spin-coated sample substrate was placed on a quartz boat and loaded in a quartz tube furnace (Model ARF-50KC, Asahi Rika Seisakusho Co. Ltd., Chiba, Japan) then heat-treated in air at 600 °C for 20 h with a heating/cooling rate of 20 °C·min^−1^. The sample was labelled CoSiOAlcoat.

#### 2.1.2. Co-Doped SiO_2_/γ-Al_2_O_3_ Composite Powder Sample 

γ-Al_2_O_3_ powder was prepared from γ-AlOOH sol using a previously reported procedure [28]. γ-Al_2_O_3_ powder was mixed with Co-doped SiO_2_ sol with Co/Si = 1/2 at the Al:Si:Co ratio of 85:10:5. The resulting mixture was heated with stirring to 80 °C and kept at this temperature overnight to give a dried Co-doped SiO_2_ gel/γ-Al_2_O_3_ powder sample. The dried powders were placed on a quartz boat and set in a quartz tube furnace (Model ARF-50KC, Asahi Rika Seisakusho Co. Ltd., Chiba, Japan) then heat-treated in air at 600 °C for 20 h with a heating/cooling rate of 20 °C·min^−1^. The sample was labelled CoSiOAlpow. 

#### 2.1.3. Homogeneous Co-Doped Aluminosilicate Powder Sample

To study the redox behavior of the Co cation in an amorphous aluminosilicate matrix, Co-doped aluminosilicate sol was prepared according to an atomic ratio of Al (in Al(NO_3_)_3_·9H_2_O):Si (in Si(OC_2_H_5_)_4_):Co (in CoNO_3_·6H_2_O) = 8:8:1. A 50 mL round-bottom flask equipped with a magnetic stirrer was charged with Co(NO_3_)_2_·6H_2_O (0.343 g), Al(NO_3_)_3_·9H_2_O (3.262 g), and C_2_H_5_OH (20 mL). To this solution, 1.92 mL (8 mmol) of Si(OC_2_H_5_)_4_ and H_2_O_2_ (2.3 mL, 30% aq) were added and maintained at RT with stirring for 1 h, then heated at 60 °C for 2 h to give a Co-doped aluminosilicate sol. Then, the resulting sol was dried at 80 °C overnight to give the dried gel powder sample. The samples were placed on a quartz boat in a quartz tube furnace (Model ARF-50KC, Asahi Rika Seisakusho Co. Ltd., Chiba, Japan) and heat-treated in air at 600 °C for 3 h with a heating/cooling rate of 20 °C·min^−1^. This sample was labelled CoSiOAlpow2.

As reference data, the textural properties were characterized for the composite powder samples. The surface morphology and the pore size distribution (PSD) of the CoSiOAlpow and CoSiOAlpow2 samples are shown in Appendix A, respectively. The CoSiOAlpow sample kept the morphology of the γ-Al_2_O_3_ polycrystallites (Appendix A), while the CoSiOAlpow2 sample exhibited a featureless glassy surface (Appendix A). The N_2_ adsorption and desorption isotherms of the CoSiOAlpow sample exhibited Type-IV isotherms according to the IUPAC classifications [29] and formed a hysteresis loop indicating the presence of mesoporosity, while those of CoSiOAlpow2 sample presented Type-II isotherms [29] without a distinct hysteresis loop (Appendix A). The PSD curve characterized by the BJH method [30] for each powder sample was consistent with the N_2_ adsorption-desorption behavior: The CoSiOAlpow sample exhibited a PSD curve with a dominant peak at 8 nm, while the CoSiOAlpow2 sample showed a broad and weak peak ranging from 20 to 100 nm (Appendix A). The resulting Brunauer–Emmett–Teller (BET) surface areas of the CoSiOAlpow and CoSiOAlpow2 samples were measured to be 249 m^2^g^–1^ and 94 m^2^g^–1^, respectively.

#### 2.1.4. γ-Al_2_O_3_/Co-Doped SiO_2_/γ-Al_2_O_3_ Layered Composite Membrane Sample

A mesoporous γ-Al_2_O_3_ layer was formed on an outer surface of a macroporous α-Al_2_O_3_ support (tubular-type, 6 mm outer diameter, 2 mm thickness, and 60 mm length, porosity 40%, Noritake Co., Ltd., Aichi, Japan) according to a published procedure [28]. The mesoporous γ-Al_2_O_3_-modified tubular support was dipped into Co-doped SiO_2_ sol with Co/Si = 1/4 at 10 mm·s^−1^, kept in the sol for 30 s, then pulled out at 1 mm·s^−1^ before heat treatment at 600 °C for 20 h under the same manner as that for the CoSiOAlpow sample synthesis. This dip-coating-heat treatment sequence was repeated once to form a Co-doped SiO_2_ thin film on the supported mesoporous γ-Al_2_O_3_ layer. Then, the Co-doped SiO_2_ thin film was coated with γ-AlOOH sol (diluted as 0.5 mol·L^−1^) and heat-treated in air at 600 °C for 3 h by using the same manner as described above to give γ-Al_2_O_3_/Co-doped SiO_2_/γ-Al_2_O_3_ three-layered composite membrane. The sample was labelled CoSiOAlmemb.

### 2.2. Characterizations

#### 2.2.1. Characterization Techniques

X-ray diffraction (XRD) measurements were performed on powder samples (Model X’pert Pro 1, Philips Ltd., Amsterdam, The Nederlands).

X-ray photoelectron spectroscopic (XPS) analysis was performed on the CoSiOAlpow2 sample with an Al K*α* X-ray source operated at 14 kV and 14 mA (Model PHI-5000, Ulvac-phi, Kanagawa, Japan). An alignment on the C 1s peak (at 284.8 eV) was performed before survey scans. To investigate the reversible redox property of the Co cation in the aluminosilicate, the CoSiOAlpow2 sample was heat-treated at 350 °C for 1 h with a heating/cooling rate of 5 °C·min^−1^ under H_2_ flow. Then, under Ar flow, the sample was heat-treated at 350 °C for 10 h with a heating/cooling rate of 5 °C·min^−1^. The heat treatments were cyclically performed under flowing of H_2_–Ar–H_2_ or H_2_–Ar–H_2_–Ar using a catalyst analyzer (BELCAT-A, MicrotracBEL Corp., Osaka, Japan). Then, Co 2p spectrum was recorded for the heat-treated sample. 

To investigate the distribution of Co cations within the CoSiOAlcoat sample, the depth-profiling XPS analysis was carried out by combining Ar ion gun etching cycles at 3 kV. The etching rate was calibrated as 3.73 nm min^−1^ by measuring that of the SiO_2_ substrate.

Surface morphology of CoSiOAlpow and CoSiOAlpow2 samples and the cross-section of the CoSiOAlmemb sample were observed by a scanning electron microscope (SEM, model JSM-6010LA, JEOL Ltd., Tokyo, Japan) operated at a voltage of 15 kV.

The BET surface area of the CoSiOAlpow and CoSiOAlpow2 samples was evaluated by measuring N_2_ adsorption and desorption isotherms at −196 °C under relative pressure ranging from 0 to 0.99 (Model Belsorp Max, BEL Japan Inc., Osaka, Japan). 

Nanostructure of the CoSiOAlpow sample was characterized by transmission electron microscopy (TEM) using a high-angle annular dark-field-scanning transmission electron microscope (HAADF-STEM, Model JEM-ARM200F, JEOL Ltd., Tokyo, Japan, operated at an accelerating voltage of 200 kV). The distribution of the constituent elements of Si, Al, Co, and O was measured and analyzed by the energy dispersive X-ray spectrometer (EDS, Model JED-2300, JEOL Ltd., Tokyo, Japan) mounted on a JEM-ARM200F. 

Local structure of γ-Al_2_O_3_ powder sample was studied by ^27^Al solid-state magic angle spinning nuclear magnetic resonance (MAS NMR) spectroscopic analysis with a 600 MHz NMR spectrometer (Model JNM-ECA600II, JEOL Ltd., Tokyo, Japan) operating at a static magnetic field of 14.01 T (155.4 MHz). Single gas permeances of the CoSiOAlmemb sample were evaluated for He and H_2_ by the constant-volume manometric method [31]. The evaluations were performed at 100–500 °C according to the procedure as reported elsewhere [32,33]. The pressure on the gas feed side in this study was maintained at 120 kPa. On the gas permeate side, the gas line and the buffer tank with the total volume of 3.11 × 10^−4^·m^3^ were completely evacuated by using an oil rotary vacuum pump. After the evacuation was terminated, the pressure increase rate at the inside of the buffer tank was measured three times. The average of the three measurements was used for calculation of the permeance of gas-*i* (*Q_i_*). 

#### 2.2.2. In Situ Characterizations 

The hydrogenation/dehydrogenation reactions were monitored in situ for the CoSiOAlpow sample by diffuse reflectance infrared Fourier transform spectroscopy (DRIFTS, Model Spectrum 100, Perkin Elmer, Waltham, MA, USA). The Co-doped SiO_2_/γ-Al_2_O_3_ composite powder sample was loaded in a diffuse reflection cell (Model STJ900C Diffuse IR Heat Cham, S.T. JAPAN Inc., Tokyo, Japan), and IR spectrum was recorded after each treatment: heat-treatment to remove adsorbed water at 500 °C for 8 h under a flow of Ar (4 mL·min^−1^).subsequent heat treatment at 500 °C for 0.5 h under a flow of H_2_ (4 mL·min^−1^).final heat treatment at 500 °C for 8 h under a flow of Ar (4 mL·min^−1^).

Temperature-programmed reduction (TPR) and desorption (TPD) experiments were performed on the CoSiOAlpow sample using a catalyst analyzer (BELCAT-A, MicrotracBEL Corp., Osaka, Japan) fixed with a quadrupole mass spectrometer (BEL Mass, MicrotracBEL Corp., Osaka, Japan). The TPR profile was measured during heating from 100 °C to 500 °C (5 °C·min^−1^) under a flow of 10% H_2_/Ar (50 mL·min^−1^). Then, the sample powder was maintained under pure-H_2_ flow (50 mL·min^−1^) for 1 h and cooled down to room temperature. The subsequent TPD measurement was carried out by heating up to 500 °C (5 °C·min^−1^) under He flow (50 mL·min^−1^). This cyclic TPR/TPD measurement was repeated 3 times.

In situ X-ray adsorption fine structure (XAFS) spectroscopic analyses were conducted at the BL5S1 of Aichi Synchrotron Radiation Center (AichiSR, Aichi Science & Technology Foundation, Seto, Aichi, Japan). The experimental set up is shown in Figure 1. The absorption spectrum can be obtained based on Equation (4), where μ and *t* are absorption coefficient and sample thickness, respectively.
μ*t* = −ln(I_1_/I_0_)(4)

The CoSiOAlpow sample (3.1 mg) was diluted with hexagonal boron nitride powder (100 mg, provided by the AichiSR, Seto, Aichi, Japan) using an agate mortar and pestle, then it was uniaxially pressed into a disk with a diameter of 7 mm. The disk sample was loaded in a quartz cell equipped with an electric heater and gas lines (Model 2000-1431, Makuharikagaku Garasu Seisakusho Inc., Chiba, Chiba, Japan) and maintained for 30 min under Ar flow. Then, under H_2_ flow, the disk sample was heat-treated at 500 °C for 30 min with a heating rate of 8 °C·min^−1^. Subsequently, under Ar flow, the furnace was cooled down to RT with a cooling rate of 8 °C·min^−1^ then maintained at RT for additional 1 h. During this treatment, the spectrum was recorded stepwise at each measurement as follows: (1) at RT under initial Ar flow, (2) after H_2_ treatment at 500 °C for 30 min, (3) at RT after being cooled down from 500 °C under Ar flow, and (4) at RT after being maintained under Ar flow for 1 h. In this study, the measurements were also performed on Co_3_O_4_ (4N, Kojundo Chemical Laboratory Co., Ltd., Saitama, Japan), CoO, and Co-foil (stored at AichiSR) at RT as reference samples. Then, to investigate the reversible redox property of the CoSiOAlpow sample, X-ray absorption near edge structure (XANES) spectroscopic analyses were performed using software (ATHENA, ver. 0.9.24/Demeter [34]).

## 3. Results and Discussion

### 3.1. Distribution of Co Cations within the Co-Doped SiO_2_/γ-Al_2_O_3_ Layered Composite Sample

To investigate the chemical composition at the hetero interface between the Co-doped SiO_2_ and γ-Al_2_O_3_, a Co-doped SiO_2_ thin layer was formed on a mesoporous γ-Al_2_O_3_ layer supported on a SiO_2_ substrate. This sample labelled CoSiOAlcoat was deposited by spin-coating of the Co-doped SiO_2_ sol with Co/Si = 1/8 followed by heat treatment in air at 600 °C for 20 h as detailed in the experimental section. Figure 2 presents the results of depth-profiling XPS analysis for the constituent elements of Si, Co, and Al. 

The thickness of the Co-doped SiO_2_ top layer was approximately 37 nm, estimated based on the Ar sputtering rate determined for the SiO_2_ substrate. Then, almost all of the mesopore channels of the γ-Al_2_O_3_ layer with a thickness of about 85 nm were infiltrated by the Co-doped SiO_2_. During the spin-coating process, Co-doped SiO_2_ sol solution easily penetrated into the mesoporous γ-Al_2_O_3_ layer. However, the depth profile of Co revealed that the Co/Si ratio of the top layer was approximately 1/37, much lower than the nominal composition of Co-doped SiO_2_ coating sol (1/8). On the other hand, around the Co-doped SiO_2_/γ-Al_2_O_3_ hetero interface (around 50 nm in depth), the Co content increased to approximately 3 at.%, then, within the mesoporous γ-Al_2_O_3_ layer at 60–100 nm in depth, the Co content was shown to be constant at a value of ca. 2 at.% and the resulting Al:Si:Co ratio was 85:10:5.

The high Co concentration at the interface of the γ-Al_2_O_3_ layer was consistent with the previous report on the synthesis of Co-doped SiO_2_ membrane on a porous Al_2_O_3_ support by Diniz Da Costa et al. [14]. The observed high concentration of Co around the hetero interface revealed preferential diffusion of Co cations from the SiO_2_ matrix to the γ-Al_2_O_3_ surface. As shown in Appendix A, ^27^Al solid-state MAS-NMR spectroscopic analysis reveals that the γ-Al_2_O_3_ prepared in this study was composed of AlO_4_ and AlO_6_ units with the AlO_6_/AlO_4_ unit ratio of about 2.4, which is consistent with reported data [35]. Moreover, our previous study on mesoporous γ-Al_2_O_3_ revealed that a considerable amount of the Al–OH group existed at the γ-Al_2_O_3_ surface, which led to the highly hydrophilic property of γ-Al_2_O_3_ [28]. Accordingly, the driving force for the Co diffusion is due to the charge compensation for the AlO_4_ site within the amorphous aluminosilicate formed in situ at the hetero interface, m[(SiO_4_)(Al^−^O_4_)]Co^m+^ (m = 2 or 3). Based on these results, the Al:Si:Co ratio of 85:10:5 was selected as an equilibrium chemical composition to investigate the redox property of Co cations at the SiO_2_/γ-Al_2_O_3_ hetero interface. 

### 3.2. Properties of Co-Doped SiO_2_/γ-Al_2_O_3_ Composites

To investigate the later redox properties of Co cations at the SiO_2_/γ-Al_2_O_3_ hetero interface, we prepared powders labelled CoSiOAlpow with the equilibrium chemical composition (Al:Si:Co = 85:10:5). We first investigated the structural and microstructural changes of the material after H_2_ treatment (the reductive condition) at 500 °C. As shown in Figure 3, the X-ray diffraction pattern of the CoSiOAlpow sample is composed of XRD peaks of γ-Al_2_O_3_, indicating that the Co-doped SiO_2_ layer is X-ray amorphous. Then, after the H_2_ treatment at 500 °C, it retains the X-ray amorphous state of the Co-doped SiO_2_ phase.

TEM observation of the 500 °C H_2_-treated CoSiOAlpow sample (Figure 4) also resulted in the detection of γ-Al_2_O_3_ as a single crystalline phase by the selected area electron diffraction ring pattern analysis (inset in Figure 4a), and the composite powder sample was composed of polycrystalline γ-Al_2_O_3_ of several nanometers in size (Figure 4b). Then, STEM-EDS mapping for the constituent elements of Al, Si, Co, and O within the area of the annular dark-field (ADF)-STEM image (Figure 5a) revealed homogeneous distribution of Co and Si as well as Al and O (Figure 5b–e). Accordingly, the composite sample powders were characterized as γ-Al_2_O_3_ polycrystalline homogeneously modified with Co-doped amorphous SiO_2_.

### 3.3. Redox Behavior of Co Species 

Redox properties of the Co cation in the CoSiOAlpow sample were assessed by in situ using XAFS spectroscopic analyses. The CoSiOAlpow sample was heat-treated at 500 °C under H_2_ flow followed by cooling to RT under Ar flow (Figure 6a). The spectra recorded at the measurement points shown in Figure 6a are shown in Figure 6b.

The CoSiOAlpow sample at RT under the initial Ar flow (Ar(0) in Figure 6a) exhibited a spectrum with a peak at 7726 eV, and this spectrum was different from those of references, Co_3_O_4_, CoO, and Co-foil recorded at RT. The unique behavior is that the peak intensity at 7726 eV changes reversibly (Figure 6c): After the 500 °C H_2_-treatment (H_2_ in Figure 6a), the peak intensity decreases, i.e., approaches the intensity level of the Co-foil at 7726 eV, then increases in a stepwise manner under Ar flow after cooling down to RT (Ar(1) in Figure 6a) followed by being maintained at RT for 1 h (Ar(2) in Figure 6a).

As discussed above, the metallic Co was not detected by our intensive HRTEM-STEM analyses. Thus, the spectra shown in Figure 6b suggested that the Co species in the CoSiOAlpow sample were Co(III) and Co(II) with the Co(III)/Co(II) ratio lower than that in Co_3_O_4_ (Co(III)/Co(II) = 2/1). Moreover, the peak intensity change shown in Figure 6c suggests the reversible redox property of the Co cation species in the CoSiOAlpow sample under the present cyclic condition of Ar (RT)-H_2_ (500 °C)-Ar (RT) treatment. 

To investigate the nature of the Co-doped SiO_2_/γ-Al_2_O_3_ hetero interface during the redox reactions of Co species, in situ DRIFT spectroscopic analysis was performed on the CoSiOAlpow sample under a cyclic heat-treatment condition under Ar flow, followed by 10% H_2_/Ar flow and subsequent Ar flow at 500 °C. The results are shown in Appendix A. Due to the heterogeneous structure with the Co-doped SiO_2_/γ-Al_2_O_3_ hetero interface, the CoSiOAlpow sample exhibited broad spectra of OH groups in the range of 3800–3200 cm^−1^: The initial DRIFT spectrum under Ar flow at 500 °C exhibited a broad peak around 3736 cm^−1^ attributed to free Si–OH groups [36,37] together with a broader one from 3640–3300 cm^−1^ assigned to hydrogen-bonded X–OH (X = Si or Al) group [23,38] (Appendix A). After heat treatment under 10% H_2_/Ar flow for 5 h at 500 °C, the intensity of the broader peak at 3640–3300 cm^−1^ increased (Appendix A), then decreased to be close to the initial intensity level after the subsequent Ar treatment at 500 °C (Appendix A). Accordingly, it is suggested that the reduction-oxidation of Co cation species in the CoSiOAlpow sample is associated with OH formation-deformation, i.e., hydrogenation-dehydrogenation.

To identity the hydrogenation-dehydrogenation reactions, cyclic TPR/TPD analysis was performed on the CoSiOAlpow sample. In this attempt, the first TPR/TPD was conducted to initialize the oxidation state of the composite powder sample, then the TPR/TPD profiles were recorded for the 2nd and 3rd cycles. As shown in Appendix A, both the 2nd and 3rd TPR profiles showed a weak broad peak for H_2_ up-take at 250–300 °C, and a larger one at 400–500 °C, then the H_2_-uptake continued to some extent under the isothermal H_2_-treatment at 500 °C. Both the 2nd and 3rd TPD profiles showed appropriate desorption peaks, a very weak peak at 250–350 °C, and another larger one during the isothermal Ar-treatment at 500 °C. The simultaneous MS analysis resulted in the detection of a trace amount of H_2_ as a single desorption component at the two temperature regions appropriate to the desorption peaks. 

These results suggest the following: The reduction-oxidation of Co cation species in the CoSiOAlpow sample is associated with the formation-deformation of OH groups.The redox reactions begin to proceed at relatively low temperatures around 250–350 °C.The redox reactions are reversible and governed by the H_2_ partial pressure at *T* ≥ 250 °C.

For further investigation on the redox reactions of the Co cation species that proceeded around 250–350 °C, we prepared another powder sample labelled CoSiOAlpow2 with homogeneous structure having almost same Co content but a higher Co/Al of 1/8 compared with that of CoSiOAlpow (1/17).

The redox behaviors of the Co cation species were investigated by cyclic heat treatment up to 350 °C. The CoSiOAlpow2 sample kept its X-ray amorphous state after the cyclic heat treatment at 350 °C under flow of H_2_–Ar–H_2_ or H_2_–Ar–H_2_–Ar (Figure 7A). As shown in Figure 7B, the XPS spectrum of as-synthesized CoSiOAlpow2 sample (Green line) exhibited a broad peak centered at 782 eV attributed to Co^2+^ together with another broad peak around 788 eV corresponding to a Co^2+^ satellite peak (labeled * in the graph) [39]. In addition, on the right side of the Co^2+^ peak, there was a peak shoulder at 779.9 eV assigned to Co^3+^ [39]. The Co^3+^ peak intensity at 779.9 eV apparently decreased after the second H_2_ treatment of the H_2_–Ar–H_2_ run (blue line), while that due to metallic Co (777.8 eV) [39] was not observed. Then, after the subsequent Ar treatment (H_2_–Ar–H_2_–Ar run), the Co^3+^ peak intensity increased to some extent (orange line).

The oxidation state of the Co cation in the starting Co(NO_3_)_2_·6H_2_O was divalent (II). Moreover, metallic Co was not formed under the present heat treatment conditions, and thus formation of Co(III) via the disproportionation reaction (3Co^2+^ → 2Co^3+^ + Co) was excluded. It should be also noted that the oxidation state of the Co cation in the polymer-derived Co-doped amorphous SiO_2_ was completely divalent [40]. 

It is well known that the Fenton reaction is the Fe(III/II)-catalyzed oxidation of organic compounds (RH) via generation of highly reactive radical species (•OH) in the presence of H_2_O_2_ [41,42]:Fe^2+^(aq) + H_2_O_2_ → Fe^3+^(aq) + OH^−^ + •OH(5)
RH + •OH → •R + H_2_O(6)

Recently, Ling reported [43] that Co(II) analogously generates •OH in the presence of H_2_O_2_: Co^2+^(aq) + H_2_O_2_ → Co^3+^(aq) + OH^−^ + •OH(7)

In this study, H_2_O_2_ was used to prepare the precursor solutions for the Co-doped SiO_2_ system, which resulted in the partial formation of Co(III) cations within the sample powders via the Fenton reaction. The results obtained in the present study reveal that reversible Co(III)/Co(II) redox reactions are associated with OH formation-deformation, which is governed by the H_2_ partial pressure at 250–500 °C:(8)$${2(\mathrm{Si}–O–\mathrm{Al})}_{n}–O^{-}\;+\;\overset{+\;H_{2}}{\underset{-H_{2}\;\mathrm{in}\;\mathrm{Ar}}{⇌}}\;{2\mathrm{Co}}^{3+}\cdot {2(\mathrm{Si}–O–\mathrm{Al})}_{n}–\mathrm{OH}\;+\;{2\mathrm{Co}}^{2+}$$

The H_2_ up-take/desorption of the CoSiOAlpow sample was mainly detected at higher temperatures around 500 °C (Appendix A). This could be explained by the kinetic factor necessary for H_2_ diffusion through the amorphous SiO_2_ network between the top surface and the SiO_2_/γ-Al_2_O_3_ hetero interface where Co cations mainly existed.

### 3.4. Gas Permeation Properties of γ-Al_2_O_3_/Co-doped SiO_2_/γ-Al_2_O_3_ Layered Composite Membrane

The cross-sectional SEM image of the layered composite membrane sample labelled CoSiOAlmemb is shown in Figure 8. In this study, the initial Co content of the SiO_2_ layer was reduced from Co/Si = 1/2 to 1/4, then the Co-doped SiO_2_ layer (Co/Si = 1/4) was placed between the two mesoporous γ-Al_2_O_3_ layers to investigate the relationship between the Co(III)/(II) redox reactions at the SiO_2_/γ-Al_2_O_3_ hetero interface and gas permeation properties. Initially, He permeance was measured at 100–500 °C. As shown in Figure 9a, initial He permeance (*Q*_He_-1st) at 100 °C was 3.1 × 10^−6^ mol·m^−2^·s^−1^·Pa^−1^, then it slightly increased with the permeation temperature. On other hand, after H_2_-treatment at 500 °C, the H_2_ permeance (*Q*_H2_) at 500 °C was slightly higher than that of *Q*_He_-1st, and the *Q*_H2_ showed an opposite temperature dependency, with decreasing temperature, the *Q*_H2_ increased. Finally, after the subsequent He-treatment at 500 °C for 8 h, the He permeance (*Q*_He_-2nd) at 500 °C was close to that of *Q*_He_-1st, then showed a positive temperature dependency similar to that observed for *Q*_He_-1st.

The temperature dependency of the *Q*_He_-1st and *Q*_He_-2nd reveals that He dominantly permeates through micropore channels with a pore size close to the kinetic diameter of He (0.26 nm) [44], and both *Q*_He_-1st and *Q*_He_-2nd are governed by activated diffusion, theoretically expressed as,
(9)$$Q_{i}=Q_{0}\exp\left(-\frac{Ea}{RT}\right)$$
where *Q_i_* is the permeance of permeate gas-i, *Q*_0_ is the pre-exponential factor (mol·m^−2^·s^−1^·Pa^−1^), *Ea* is the activation energy (J·mol^−1^), *R* is the gas constant (8.314 J·mol^−1^·K^−1^), and *T* is the temperature (K) [45,46].

On the other hand, as shown in Figure 9a, the *Q*_H2_ is proportional to the inverse square root of permeation temperature (*T*^−0.5^), and is thus suggested to be governed by the Knudsen’s diffusion mechanism. The theoretical gas permeance based on the Knudsen diffusion through a membrane having pore radius (*d**_p_*), porosity (*ε*), tortuosity (*τ*), and thickness (*L*) is formulated using the molecular weight of the permeating gas-i (*M_i_*) and the permeation temperature (*T*) as follows,
(10)$$Q_{i}=\frac{\varepsilon_{i}d_{p,i}\rho_{i}}{\tau_{i}L_{i}}\sqrt{\frac{8}{\pi M_{i}RT}}$$
where *ρ* is the probability of diffusion through the pore channel, *ρ_i_*, *ε_i_*, *dp_i_*, *τ_i_*, and *L_i_* are membrane structural factors due to possible dependence on the size of permeate gas-i [47]. 

Knudsen diffusion occurs when the mean free path of the permeate gas molecule is relatively longer than the pore diameter, so the permeate gas molecules collide frequently with the pore wall. Generally, Knudsen diffusion is dominant for mesopores (2–50 nm) [28,47]. Since the kinetic diameter of He (0.26 nm) is smaller than that of H_2_ (0.289 nm) [44], prior to the activated diffusion through micropores, He molecules should exhibit Knudsen diffusion characteristics by permeating through the mesopore channels where H_2_ molecules dominantly permeate. These results suggest that the size of the gas permeable channel pores alternatively change from micropores under He flow to mesopores under H_2_ flow, then back to micropores under He flow.

Time dependence of the ***Q*_He_** was monitored after maintaining the CoSiOAlmemb sample under H_2_ flow for 4 h. In this attempt, H_2_ treatment and subsequent *Q*_He_ measurements were performed at the lower temperature of 300 °C to highlight the time dependence. As shown in Figure 9b, the initial *Q*_He_ of 3.53 × 10^−6^ mol·m^−2^·s^−1^·Pa^−1^ slightly decreased with time and reached 3.45 × 10^−6^ mol·m^−2^·s^−1^·Pa^−1^ after 24 h. Then, after the second H_2_ treatment for 4 h, the *Q*_He_ regained close to its initial level. Accordingly, the size of pore channels under the H_2_ flow became larger, and then reduced to be close to the initial size. This reversible change in pore size could be explained by the reversible redox of Co(III)/(II) associated with the OH group formation/deformation (Equation (8)), i.e., bond cleavage-regeneration of the Si–O–Al amorphous network governed by the H_2_ partial pressure at 300–500 °C.

## 4. Conclusions

In this study, Co-doped SiO_2_/γ-Al_2_O_3_ composite samples were synthesized by conventional sol–gel methods using H_2_O_2_. The redox behaviors of the Co cations in the composite samples were studied and the results can be summarized as follows:XPS depth profile analyses for the Co-doped SiO_2_/mesoporous γ-Al_2_O_3_ layered composite sample revealed preferential Co diffusion from the SiO_2_ top-layer matrix to the γ-Al_2_O_3_ surface, and the equilibrium chemical composition of the Al, Si, and Co within the mesoporous γ-Al_2_O_3_ layer was determined as Al:Si:Co = 85:10:5.XRD, HRTEM, and HAADF-STEM analyses revealed that the Co-doped SiO_2_/γ-Al_2_O_3_ composite powder sample with the equilibrium composition of Al:Si:Co = 85:10:5 kept its amorphous state without crystallization of Co oxides after H_2_ treatment at 500 °C.In situ XANES spectroscopic analyses during the 500 °C heat treatment under H_2_ flow and subsequent cooling to RT under Ar flow exhibited reversible redox properties of Co cations in the Co-doped SiO_2_/γ-Al_2_O_3_ composite powder sample.XPS and DRIFT spectroscopic analyses and cyclic TPR/TPD measurements concluded the reversible Co(III)/Co(II) redox reactions were associated with OH formation (hydrogenation)-deformation (dehydrogenation) within amorphous aluminosilicate, which was governed by the H_2_ partial pressure at 250–500 °C.The Co(III) cations in the present composite samples formed via the Fenton reaction in the presence of H_2_O_2_ during the sample syntheses.Gas permeation measurements for the γ-Al_2_O_3_/Co-doped SiO_2_/γ-Al_2_O_3_ layered composite membrane under the cyclic He–H_2_–He flow suggested that H_2_ triggered a chemical valve property: micropores under He flow (closure) and mesopores under H_2_ flow (open), which could be explained by the reversible redox reactions of Co(III)/(II) associated with bond cleavage (hydrogenation)-regeneration (dehydrogenation) of the Si–O–Al amorphous network formed in situ at the hetero interface of the Co-doped SiO_2_/γ-Al_2_O_3_.

The observed H_2_-triggered chemical valve response needs a long time to function, especially for closure that takes approximately 8 h at 500 °C under Ar flow. The relations between the relative amount of Co(III) in the Si–O–Al amorphous network, recyclability of the redox reactions, and the valve response remain as future subjects.

## Figures and Tables

**Figure 1 materials-13-05345-f001:**
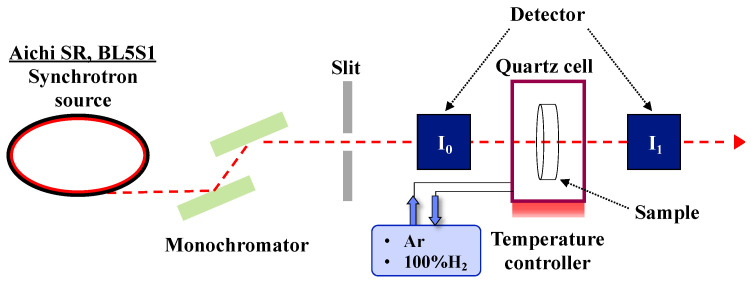
Experimental set up for in situ XAFS spectroscopic analysis.

**Figure 2 materials-13-05345-f002:**
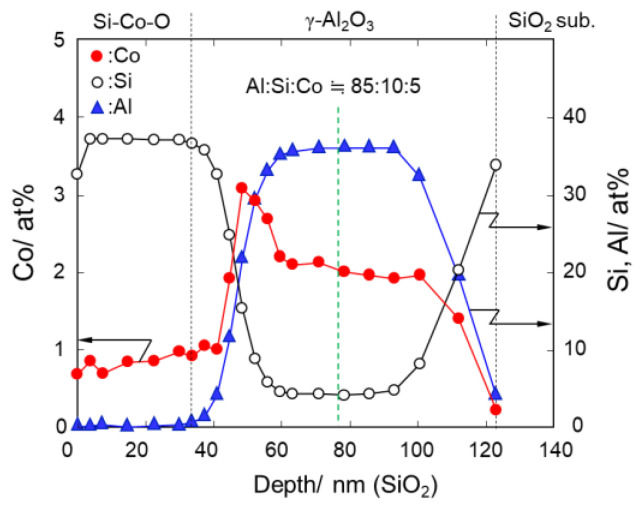
Depth-profiling XPS analysis for Co, Si, and Al within the CoSiOAlcoat sample.

**Figure 3 materials-13-05345-f003:**
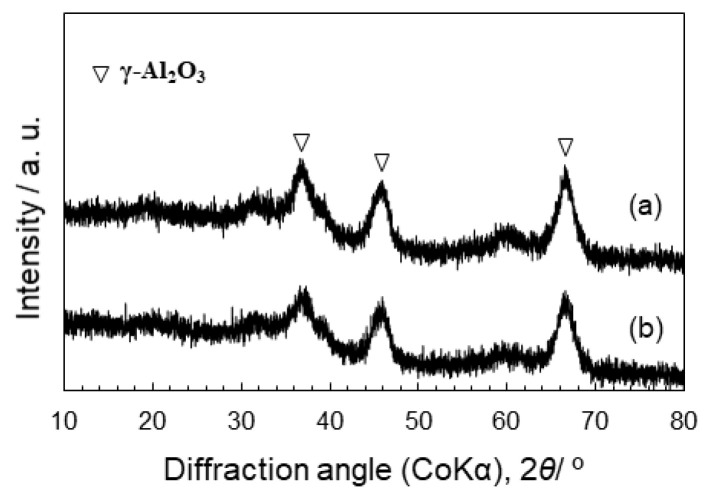
XRD patterns of the CoSiOAlpow sample. (**a**) As synthesized and (**b**) after H_2_ treatment at 500 °C for 1 h.

**Figure 4 materials-13-05345-f004:**
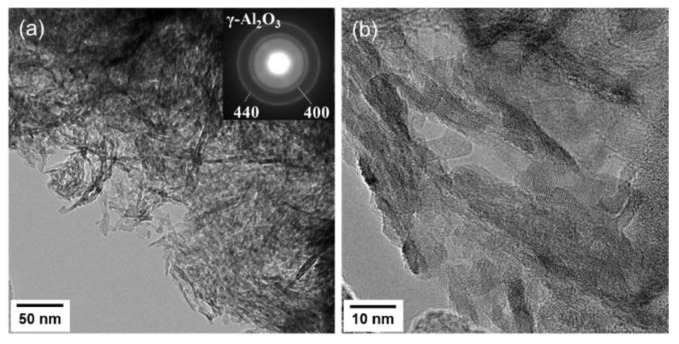
(**a**) TEM image of the CoSiOAlpow sample and the corresponding selected area electron diffraction pattern obtained, and (**b**) HRTEM image showing γ-Al_2_O_3_ polycrystalline of several nanometers in size.

**Figure 5 materials-13-05345-f005:**
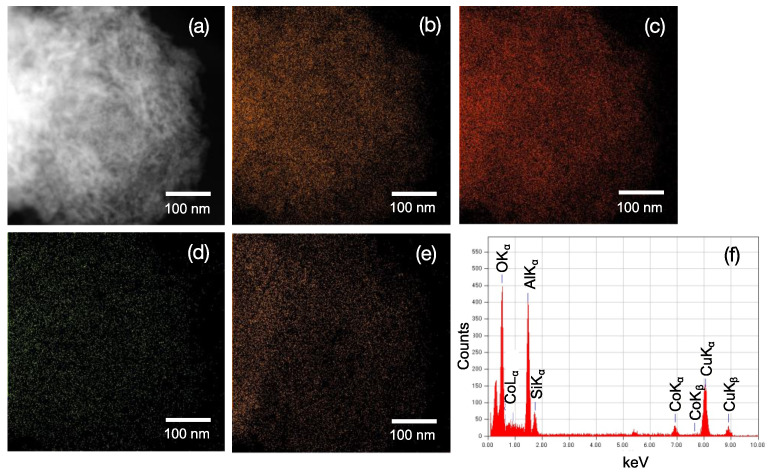
STEM-EDS mapping analysis for the CoSiOAlpow sample. (**a**) Annular dark field (ADF)-STEM image and the results of EDS mapping of (**b**) Al, (**c**) O, (**d**) Co, and (**e**) Si. (**f**) Typical EDS spectra recorded for the mapping analyses.

**Figure 6 materials-13-05345-f006:**
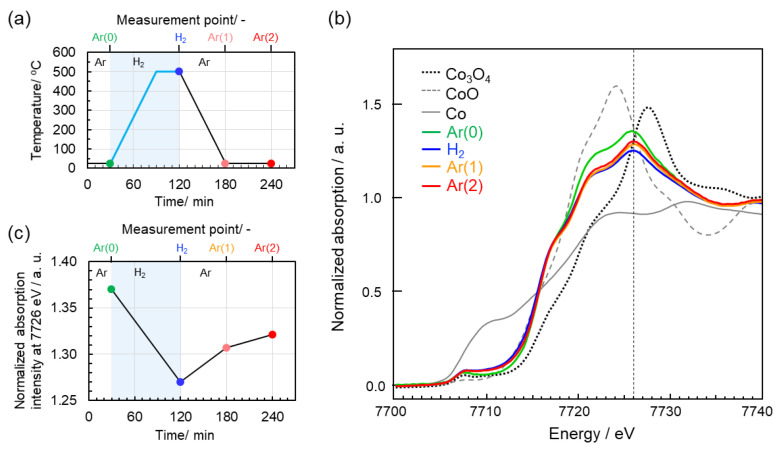
Results of in situ XAFS spectroscopic analysis for the CoSiOAlpow sample. (**a**) Heat treatment conditions and spectrum measurement points in this study, (**b**) XANES spectra of the CoSiOAlpow and reference samples, and (**c**) change in normalized absorption peak intensity at 7726 eV shown in (**b**).

**Figure 7 materials-13-05345-f007:**
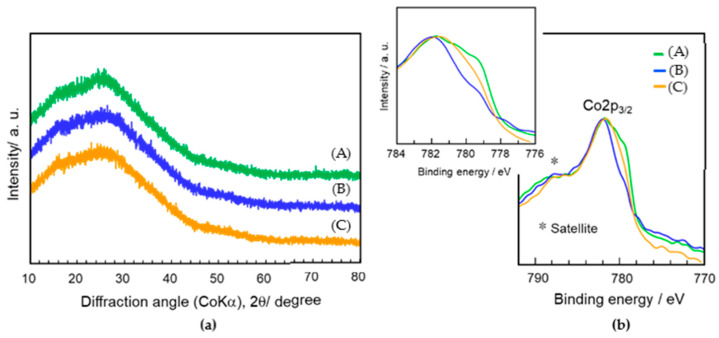
(**a**) XRD patterns and (**b**) XPS spectra of the CoSiOAlpow2 sample. (**a**) As-synthesized, after (**b**) heat treatment at 350 °C for 1 h under H_2_ flow and the subsequent heat treatment at 350 °C for 10 h under Ar flow.

**Figure 8 materials-13-05345-f008:**
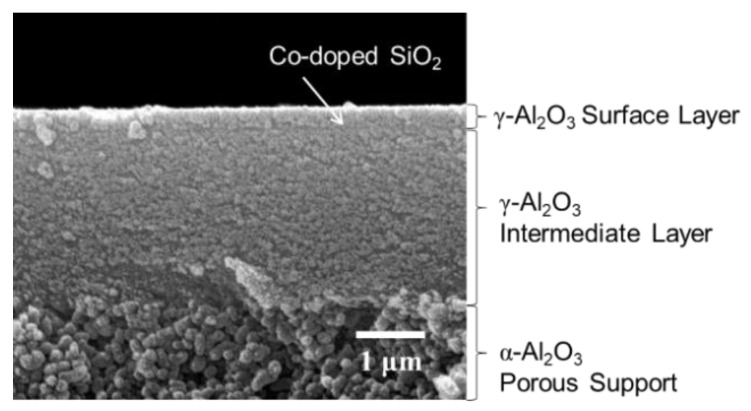
Cross-sectional SEM image of the CoSiOAlmemb sample.

**Figure 9 materials-13-05345-f009:**
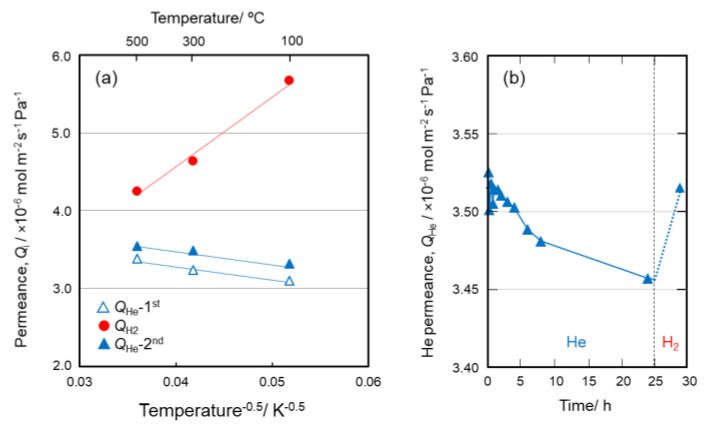
(**a**) Single gas permeation behaviors through the CoSiOAlmemb sample, and (**b**) time dependence of *Q*_He_ after H_2_ treatment at 300 °C for 4 h.

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
