# Peer review of "Reversible Redox Property of Co(III) in Amorphous Co-doped SiO2/γ-Al2O3 Layered Composites"

_materials, 2020, doi:10.3390/ma13235345_

Round 1

Reviewer 1 Report

In this study, the Co-doped SiO2/γ-Al2O3 composite materials were synthesized and the redox properties of Co cations was investigated by cyclic heart treatment in interval 300-500 °C, under alternative flow gas change (H2 and Ar). Thus, the materials were characterized highlighting their properties. Beside this, the authors obtained layered composite membrane, γ-Al2O3/Co-doped SiO2/γ-Al2O3, which were subjected to single gas permeation test, by using He and H2. The submitted manuscript is interesting, original and within the scope of the journal, but some changes should be addressed:

  1. In the introduction (lines 34-35), the authors highlighted the importance of silica (SiO2) membranes in comparison with polymer membranes. Please update the state of the art (the given references are older). It will be also interesting to compare the thermal stability of silica membranes with the mixed matrix membranes with silica (non- and functionalized) filler (please see: https://doi.org/10.3389/fchem.2019.00332).
  2. Regarding the comment from line 35 to line 37 ("The high..") and according to IUPAC nomenclature, the pore with 3 nm in size are considers mesoporous and not microporous (please see: doi:10.1351/pac199466081739). Please revise the notion.
  3. The figure titles (2, 4, 5, 6, 7, 9) are too long. Try to short them by insertion of the information in the text.
  4. Please improve the resolution for figure 1.
  5. I recommend to increase the fonts from figure 4 a.
  6. From TEM analysis the pores diameter must be extracted. Please give the interval of pores sizes in order to reveal the nature of the material (micro or mesoporous).
  7. Please present the specific surface for the powders and also the pore distribution.
  8. Why did the permeation studies have been performed at 120 kPa gas feed pressure? Beside the temperature, it will be interesting to see how the pressure variation influence the permeance.
  9. Please correct references 18 and 25 because the background is grey highlighted.

Author Response

Dear Reviewer 1:

Thank you for the valuable comments. We have considered all of your comments and revised our manuscript with respect to your suggestion. Your valuable suggestions and comments were carefully considered during the manuscript revision.

Please see the attachment:

All changes performed according to comments from Reviewer 1# and Reviewer 2# are highlighted using red word in the text. In addition, words rephrased according to similarity report are highlighted using blue word in the text.

Sincerely yours,

Yuji IWAMOTO

On behalf of all co-authors

Nagoya, Japan, November 17th, 2020.

Reviewer 2 Report

The subject of the manuscript is interesting and the authors performed a lot of experiments using advanced characterization techniques. However, before being accepted for publication the authors should improve their manuscript by considering the following aspects:

1)In the introduction section they should highlight more the practical importance of their study, as well as the economic advantages brought by the utilization of this type of membrane (CoSiOAlmemb)

line 64 - "heart" - replace by heat

2)In the experimental section

line 75 - ratio Co/Si = 1/8, 4/1, and 1/2 I think that it is 1/4 instead of 4/1

line 76 - rephrase "according to a procedure reported elsewhere" or "according to a previously reported procedure"

line 94 and 106 - rephrase "by following the published procedure" - using the previously published procedure

line 99 -" to afford γ-Al2O3 thin layer over SiO2 substrate" -rephrase "to afford the formation of γ-Al2O3 thin layer over SiO2 substrate"

line 100, line 135 - replace "under" by using

line 141 - power - powder

lines 146-151 The authors describe the analysis performed in order to investigate the reversible redox property of Co cation in CoSiOAlpow2 when the treatment with H2 was performed up to 350oC using Ar as inert gas

Meanwhile at lines 182-189 it is described that in the case of the sample CoSiOAlpow the TPR was performed from 100 up to 500oC using H2/Ar mixture, then the sample was maintained 1 h under H2 flow ? was it pure H2 in this case or the same H2/Ar mixture?

Why the authors did not consider to perform also for this sample the same test as for the sample CoSiOAlpow2 ?

Section 3 Results and discussion

The authors do not mention anything about the chemical composition of the sample CoSiOAlmemb (which are the ratios Al:Si:Co in this case?)

The authors state at lines 336-339 "For further investigation on the redox reactions of Co cation species which proceeded around 250 to 350 °C, we prepared another powder sample with homogeneous structure and higher Co content labelled CoSiOAlpow2." However if one calculates the atomic content considering the preparation presented in the experimental section (ratio Al:Si:Co=8:8:1) it gives 6% Co instead of 5% which is in the case of Al:Si:Co=85:10:5) I o not think that a so little rise of the Co amount will be significant for their purpose. Also I think that the fact that this sample has equal amounts of Al and Si may have an influence on the obtained results.

line 286 - heart-treated - change to heat-treated

Summary

line 451 H2-triggard chemical - I think it is H2-triggered chemical

Author Response

Dear Reviewer 2:

Thank you for the valuable comments. We have considered all of your comments and revised our manuscript with respect to your suggestion. Your valuable suggestions and comments were carefully considered during the manuscript revision.

Please see the attachment:

All changes performed according to comments from Reviewer 1# and Reviewer 2# are highlighted using red word in the text. In addition, words rephrased according to similarity report are highlighted using blue word in the text.

Sincerely yours,

Yuji IWAMOTO

On behalf of all co-authors

Nagoya, Japan, November 17th, 2020.

Reviewer 3 Report

In this manuscript, Tada et al. studied the reversible redox property of cobalt3+ in amorphous Co-doped SiO2/γ-Al2O3 layered composites. The author provided clear and informative introduction. Generally, the experiments in the manuscript were well designed and the data was well presented. The authors also provided detailed and clear experimental procedures in the experimental sections so that other could easily repeat their experiments. Overall, the data in this manuscript could support the conclusion the author made. I recommend the paper to be published in Materials as it.

Author Response

Dear Reviewer 3:

We are happy to lean that the manuscript has been recommend as acceptable for publication in the journal, materials.

Sincerely yours,

Yuji IWAMOTO

On behalf of all co-authors

Round 2

Reviewer 1 Report

The submitted manuscript is interesting, original and within the scope of the journal. In addition, the manuscript is well illustrated and presents in a scientific manner the subject.

I consider that the article can be recommended for the publication in present form.

Reviewer 2 Report

The authors revised their manuscript and answered to all my queries. Therefore, the manuscript can be accepted now for publication